# Cytotoxicity of Mahanimbine from Curry Leaves in Human Breast Cancer Cells (MCF-7) via Mitochondrial Apoptosis and Anti-Angiogenesis

**DOI:** 10.3390/molecules27030971

**Published:** 2022-02-01

**Authors:** Yahya Hasan Hobani

**Affiliations:** Department of Medical Laboratory Technology, Faculty of Applied Medical Sciences, Jazan University, Jazan P.O. Box 114, Saudi Arabia; yahobani@jazanu.edu.sa; Tel.: +966-542-821-130

**Keywords:** *Murraya koenigii*, curry leaves, mahanimbine, carbazole alkaloids, apoptosis, mitochondrial cell death, metastasis

## Abstract

Mahanimbine (MN) is a carbazole alkaloid present in the leaves of *Murraya koenigii,* which is an integral part of medicinal and culinary practices in Asia. In the present study, the anticancer, apoptotic and anti-invasive potential of MN has been delineated in vitro. Apoptosis cells determination was carried out utilizing the acridine orange/propidium iodide double fluorescence test. During treatment, caspase-3/7,-8, and-9 enzymes and mitochondrial membrane potentials (Δψm) were evaluated. Anti-invasive properties were tested utilizing a wound-healing scratch test. Protein and gene expression studies were used to measure Bax, Bcl2, MMP-2, and -9 levels. The results show that MN could induce apoptosis in MCF-7 cells at 14 µM concentration IC_50_. MN-induced mitochondria-mediated apoptosis, with loss in Δψm, regulation of Bcl2/Bax, and accumulation of ROS (*p* ≤ 0.05). Caspase-3/7 and -9 enzyme activity were detected in MCF-7 cells after 24 and 48 h of treatment with MN. The anti-invasive property of MN was shown by inhibition of wound healing at the dose-dependent level and significantly suppressed mRNA and protein expression on MMP-2 and -9 in MCF-7 cells treated with a sub-cytotoxic dose of MN. The overall results indicate MN is a potential therapeutic compound against breast cancer as an apoptosis inducer and anti-invasive agent.

## 1. Introduction

With approximately 1.4 million cases identified each year, breast cancer is the most prevalent disease among women, and it is the most frequently diagnosed cancer with more than 2 million new cases in 2020 according to GLOBOCAN data [1]. Genetic mutations and family history are the main cause of breast cancer [2]. Despite the availability of various treatment modalities, the rate of death and frequencies has risen worldwide in recent years. Breast cancer was responsible for 684,996 deaths worldwide, an age-adjusted mortality rate of 13.6 deaths per 100,000 people [3]. Hence this prevalence of breast cancer is a severe public health concern [4]. Because of medications that can interfere with the multiple biological processes involved in cancer cell proliferation, cytotoxic chemotherapy for breast cancer has made significant progress in recent years [5,6]. Intensive research into fundamental signaling processes involved in cell proliferation and cell death has resulted in effectively tailored cancer medicines. Regulating apoptosis is vital in creating anticancer drugs; hence, this mode of cell death has been studied abundantly. Many of the innovative small molecules and biological agents being researched target apoptosis-related pathways [7,8,9]. Reactive oxygen species (ROS) are vital for anticancer drug research because too much ROS accumulation results in DNA damage [10]. Thus, induction of ROS is a strategy of anticancer treatment. The accumulated ROS can activate initiators and executioner caspases, lose mitochondrial membrane potential (Δψ_m_), and release cytochrome c into the cytosol. All of these events lead to apoptosis [11]. Additionally, the ratio of Bcl2/Bax proteins in the mitochondria controls the tumor cells’ sensitivity to chemotherapeutic agent-induced apoptosis [12].

If detected early, breast cancer is a curable disease through surgical resection. Still, complete curing is always challenging due to its invasive and metastasis behavior to nearby organs such as bone, lung, liver, and brain, in addition to lymph glands [13]. Breast cancer cells are characterized by invasion and metastasis, which significantly diminish the life expectancy of breast cancer patients. Reduced tissue inherency, deterioration of the extracellular matrix and basement membrane structure, and greater proteolysis are factors that trigger invasion and metastasis. Accumulating evidence has shown that extracellular matrix remodeling proteinases, such as matrix metalloproteinases (MMPs), are the primary mediators of the microenvironmental changes observed during invasion and metastasis [14,15]. MMPs are endopeptidases that act on a wide variety of proteins, including gelatin, collagen, and elastin [16]. They have been investigated as possible diagnostic and prognostic indicators in various cancer types and stages. MMPs exist in six different families, in which gelatinases such as MMP-2 and MMP-9 are the essential enzymes in the invasion and dissemination of breast cancer cells [17,18].

Apart from herbs, many food ingredients have been used for centuries in treating many chronic diseases. *Murraya koenigii* is a popular condiment and spice in Asian cuisine. It is referred to as “curry leaves” and is often utilized in culinary applications due to its fragrant properties. This plant’s fresh leaves are used in practically all curry and gravy preparations for their aroma and therapeutic properties [19]. It belongs to Rutaceae family and is native to India, and is now distributed in most of southern Asia. It is widely used to cure rheumatism, traumatic injury, and snakebite [20]. This plant extract and few compounds isolated from it have been proven to have anticancer potential, particularly in breast cancer [21,22,23,24]. Studies have shown that the leaves of this plant have many other pharmacological benefits, such as immunomodulatory, anti-bacterial, anti-fungal, anti-protozoal, antioxidant, and hypolipidemic activity, etc. [19,25,26]. Even though many phytochemicals have been isolated from this plant, carbazole alkaloids show many potent pharmacological benefits. Murraya species is one of the richest sources of carbazole alkaloids [25]. A few carbazole alkaloids such as mukonal, mahanine, and girinimbine isolated from this plant have shown significant anticancer activity, particularly in breast cancer [26,27,28,29]. Mahanimbine (MN) is a carbazole alkaloid and is present in the leaves of this plant (Figure 1A). Available data show mahanimbine’s high pharmacological potential in reducing high fat-induced metabolic alterations [30], in preventing obesity [31], and for having anti-aging [32] and anti-anxiety activity [33], as well as anticancer activity in pancreatic cancer, leukemia [34], and bladder cancer [35]. Two research reports [36,37] showed that MN could inhibit MCF-7 cells significantly. Based on these reports and the earlier studies of MN as an anticancer agent, it was worthwhile to investigate its anticancer mechanism and anti-invasive properties in vitro using MCF-7 as a model.

## 2. Results

### 2.1. MN Inhibited Breast Cancer Cells Proliferation

The cell viability of breast cancer MCF-7 cells was determined using the MTT analysis to assess the anti-proliferation effects of MN on the cells. Figure 1B depicts the viability of MCF-7 cells treated with different doses of MN for 48 h. Cisplatin was used as a positive control in the assay. The MN inhibited the MCF-7 cells and MDA-MB-231 cells with an IC_50_ of 14 ± 0.2 and 21.5 ± 0.8 µM, respectively, while the IC_50_ of cisplatin was 5 ± 0.08 µM, in an agreement with an earlier study [38]. The non-cancer mammary cells (MCF-10A) had shown an IC_50_ of 30.5 ± 1.4 µM on MN treatment. Since MCF-7 cells showed the lowest IC_50_, the rest of the experiments were carried out using MCF-7 cells.

### 2.2. MN Induced Breast Cancer Cells Apoptosis

A normal inverted light microscopic examination of MN-treated MCF-7 cells after 48 h of exposure revealed typical morphological characteristics indicating apoptosis. The observed morphological alterations included decreased cell volume, cell shrinkage, and the generation of cytoplasmic blebs. Figure 2 indicates that the MCF-7 cells treated with MN 14 μM were changed into round shapes compared with untreated. On the other hand, untreated cells had a higher confluence of monolayer cells throughout the incubation time than treated cells, with lower cell volume and shrinkage.

Acridine orange and propidium iodide (AO/PI) double staining was carried out to differentiate early, late apoptosis, and necrosis cells by morphological differentiation in the MCF-7 cells treated with 14 µM of MN for 12, 24, and 48 h. Unlike PI, which only stains dead cells, acridine orange is a vital dye that stains both living and dead cells. In the MN-treated cells, early apoptosis was detected by the accumulation of acridine orange in the DNA fragments with intense green fluorescence (Figure 3B). At the same time, control cells had a complete green nuclear structure (Figure 3A). Blebbing and nuclear chromatin condensation were found after 24 h of MN treatment (Figure 3E,F). After 48 h, late apoptotic alterations such as reddish-orange staining owing to AO binding to denatured DNA were detected. Viable, early apoptotic, late apoptotic, and secondary necrosis cells were counted under a fluorescent microscope (Figure 3G). A total of 200 cells were randomly counted, along with the untreated negative control. The findings revealed that MN created morphological characteristics and a higher apoptosis signal time-dependently related to apoptosis. While performing differential scoring of treated cells, a statistically significant (*p* < 0.05) variation in the cell population was detected.

### 2.3. MN Induced Intracellular ROS

One of the critical factors that triggers apoptosis is the accumulation of ROS. To investigate the induced ROS by MN treatment, the MCF-7 cells were treated with it, and DCFDA (dichlorofluorescin diacetate) was used to measure the green fluorescence produced. As shown in Figure 4, it is evident that MN could increase the ROS. As shown in Figure 4F, at 48 h, there was a significant 2.5-fold increase in the ROS generated. The positive control used H_2_O_2_, and the accumulation of ROS was also significantly increased, which compared well with the control.

### 2.4. MN Mediated Loss of Mitochondrial Membrane Potential (Δψm)

Fluorescence microscopy of MCF-7 cells treated with MN for 12 and 24 h demonstrated a decrease in mitochondrial membrane potential with enhanced JC-1 monomers (green fluorescence) and reduced J-aggregates (red fluorescence). The MCF-7 cells treated at 0 h had hyperpolarized membrane potentials visible as a red/orange-fluorescent J-aggregate. In comparison with untreated control cells, incubating MCF-7 cells with MN resulted in a time-dependent reduction in Δψm after 12 and 48 h. The quantitative result of positive fluorescent cells revealed a one-third drop in diamer red fluorescence in 24 h, which was statistically significant (*p* < 0.05) (Figure 5). These findings indicated that MN substantially depolarized Δψm in MCF-7 cells, perhaps reflecting mitochondria-mediated cell death.

### 2.5. MN Increased Activity of Caspase-3/7 and -9 Enzymes

Caspases activity of the MN-treated cells was measured by using luminescence assay. Caspase -3/7 and -9 enzyme activity were detected in MCF-7 cells after 24 and 48 h of treatment with MN. Both caspases were considerably enhanced (*p* < 0.05) at the 14 µM concentration treatment, but caspase-8 enzymes remained unaffected throughout the treatment periods (Figure 6).

### 2.6. MN Suppresses Migration of MCF-7 cells

Endothelial cell migration is essential to angiogenesis, which enables cancer to progress in the body. The anti-migratory activity of drugs may thus prevent the progression of cancer [39]. A wound-healing assay was performed to determine the effect of MN on the anti-migratory activity in MCF-7 cells. Wound healing was photographed at the same field and magnification to compare the results between 0, 12, 24, and 48 h. This assay was performed with a maximum of 10 µM concentration of MN as sub-cytotoxic concentration, as the IC_50_ was 14 µM at 48 h. As Figure 7A shows, the wound healing was inhibited by MN at the dose-dependent level. The wound healing rate for non-treated cells (0 µM) and 2 µM treated cells was significantly faster than the other two doses, 5 and 10 µM. As shown in Figure 7B, 10 µM MN inhibited the wound closure to 80% compared with control cells, where it was closed near to complete closure.

### 2.7. Effects of MN on Apoptotic and Invasion Gene and Protein Expression in MCF-7 Cells

Real-time PCR was performed to study the mRNA expression levels of mitochondrial apoptosis and invasion-related significant genes in MCF-7 cells after treatment with MN at 12, 24, and 48 h. As shown in Figure 8A,B, there was a significant increase and decrease in the Bax and Bcl2 mRNA levels, respectively, in 14 µM MN-treated cells compared with control. At 48 h, there was 3-fold increase in the Bax (*p* < 0.05), while at the same period, a 1.5-fold decrease in Bcl2 was observed (*p* < 0.05). Meanwhile, as shown in Figure 8C,D, MN significantly suppressed mRNA expression on MMP-2 and MMP-9 in MCF-7 cells treated with 10 µM of MN. The ELISA-based protein expression study had shown that both apoptosis-related proteins (Bax and Bcl2) were regulated time-dependently. The sub cytotoxic concentration of MN reduced the MMP-2 and -9 protein expression, which was downregulated in the same levels as it happened at the translation level (Figure 9). There were time-dependent decreases in MMP, which was statistically significant at 24 and 48 h (*p* < 0.05).

## 3. Discussion

Breast cancer is one of the deadliest cancers among women, which can be cured with proper drug treatment if detected very early [40]. However, there are numerous determining side effects associated with currently available first-line breast cancer treatments, such as aberrant proliferation, a higher incidence of endometriosis, and non-specific cytotoxicity [41]. This problem is considered one of the most compelling drivers for developing novel cancer therapy options that use natural substances with the lowest cell resistance level and the fewest adverse effects. In reality, in the field of anticancer research, more than half of the total medications now in clinical trials are derived from naturally occurring compounds or their derivatives [42]. Most of these naturally derived cancer drug candidates induce their action through apoptosis. Most cancer treatment techniques, such as chemotherapy and radiation therapy, are likely to be impacted by cells’ apoptotic features; hence, it has apparent therapeutic significance. Programmed cell death is one of the desired modes of cell death in cancer drug discovery, which comprises many biochemical and morphological features such as cell shrinkage, cell disintegration into membrane-bound apoptotic particles, and phagocytosis by neighboring cells [43]. Mahanimbine (MN) is essentially a secondary metabolite that belongs to carbazoles alkaloids isolated from curry leaves. According to the current findings, MN is a cytotoxic secondary metabolite that promotes apoptosis in MCF-7 cells, as seen by the double fluorescent assay employed. The results obtained agree with the earlier research conducted with MN, which has successfully induced the apoptosis mode of cell death in pancreatic and bladder cancers [34,35].

Reactive oxygen species (ROS) are highly reactive molecules with a short half-life. There is mounting evidence that ROS and mitochondria are critical in activating apoptosis under both physiological and pathologic conditions [44]. In the current research, it has been observed that MN’s apoptotic impact on MCF-7 cells was linked to a large increase in intracellular ROS levels. ROS have a critical role in cell signaling and in the control of the significant mitochondria-mediated apoptotic pathways. Mitochondria have played a crucial part in the apoptotic process due to their capacity to directly activate the cellular apoptotic program [45]. In addition to their critical involvement in the execution phase of apoptosis, it appears that mitochondrial reactive oxygen species (ROS) may contribute to cell death. Even though it is not very clear how ROS is initially released from mitochondria, the development of mitochondrial pores and reduction in its potential has been well studied as evidence of the relationship between ROS and mitochondria in apoptosis [46]. Hence, the reduction in Δψm has been studied after treatment with MN in MCF-7 cells. We observed a reduction in Δψm, indicating that that MN may operate on mitochondria, inducing Δψm depletion and subsequent apoptosis, probably via ROS. These results were in good agreement with the earlier studies carried out with *Murraya koenigii extracts in* Dalton’s Ascites Lymphoma (DL) cell line [47], *isolated carbazole alkaloids in HL-60 cells* [48], *and* Koenimbine in HepG2 cells [49]. These findings together with our results suggest that this plant and its alkaloids induced cell death via mitochondria involvement. Additionally, in this study, it was found that at 14 µM, MN leads to cell death in MCF-7 cells with significant ROS, but the same concentration (14 µM) was safe in non-cancer cells. From this, it could be estimated that the ROS generated by MN could be defended by non-cancer cells, probably through the anti-oxidant defense mechanism.

The proportion of Bcl-2 and Bax in the mitochondria regulates apoptosis produced by chemical agents via the intrinsic mechanism [50]. For the effective regulation of apoptosis through mitochondria, there must be a harmony between the pro-apoptosis activity of Bax and anti-apoptosis activity of Bcl2. The translocation of bax from the cytosol into mitochondria causes mitochondrial dysfunction by creating mitochondrial membrane transition pores and a decrease in Δψm, which leads to the potentiation of drug-induced apoptosis [45]. Apoptosis produced by carbazole alkaloids is linked to increased pro-apoptotic protein levels and/or deceased antiapoptotic protein expression, indicating that the Bax/Bcl-2 ratio is an essential apoptotic measure in these chemicals. Similar results were reported earlier. For example, bismahanine, a carbazole alkaloid, exerted anticancer effects on human cervical cancer cells by high Bax/Bcl-2 ratio [51]. An et al. reported that heptaphylline, another carbazole alkaloid induced apoptosis in bladder cancer by increased Bax/Bcl-2 ratio [52]. Moreover, a few other carbazole alkaloids such as glycoborinine and murrayanine had shown a similar mode of cell death with different cancers [53,54]. Accordingly, in the current study, MN appears to function by regulating the Bax/Bcl-2 ratio, and the augmented Bax gene expression in MCF-7 cells contributes to the apoptotic impact. Meanwhile, the transcriptional-level apoptosis protein analysis had shown a similar kind of expression, which confirms the involvement of an intrinsic way to apoptosis induced by MN.

Caspases enzymes are an inevitable factor in the sequence of apoptosis. They can be broadly divided into two types: initiator and executioner caspases. Both are different in their structure and functions [55]. Even though they are assumed to function downstream of cytochrome c release, it has been demonstrated that these caspases affect the mitochondria and upstream processes of intrinsic apoptosis [56]. It has been shown that the initiator caspase-9 decouples mitochondria and increases ROS generation [57]. In the current research, MN treatment in MCF-7 cells activated caspase-3/7 and -9. Because the caspase-3 gene has been functionally deleted in MCF-7 cells, there is no caspase-3 activity present. As a result, caspase-7 activity is what was being measured in this experiment [58]. Caspase-9 is located in the intermembrane space of mitochondria and is produced in a Bcl2-inhibitable manner following permeability transition in isolated mitochondria and upon stimulation of apoptosis in cells. This released caspase-9 is then stimulating the executioner caspases such as 3 or 7 for downstream DNA damage; in this case, it must be caspase-7 [59]. Meanwhile, the activity of caspase-8 was not estimated in the given time of experiments. Earlier, other carbazole alkaloids such as pyrayafoline-D, mahanine, and murrafoline-I also showed a similar kind of caspase-8-independent apoptosis in cancer cells [48,60]. This may be highlighting the absence of extrinsic pathways in carbazole alkaloids, particularly MN-mediated cell death in MCF-7 cells.

In the current research, caspase-9 activity was significantly observed at 24 and 48 h, but not at 12 h. Similarly, the apoptosis assay carried out earlier in this research also showed this time lag in showing non-significant apoptosis cells in the first 12 h of MN treatment. In addition, the reduction in Δψm was also not observed at 12 h. However, there was a significantly higher ROS produced at 12 h. This time lag probably showed that ROS production happening at an earlier time might have triggered the observed Δψm reduction and subsequent apoptosis events. Evidence from the previous studies shows that early ROS production and mitochondrial GSSG formation preceded mitochondrial dysfunction and apoptosis [61]. Kroemer et al. had found earlier that mitochondrial ROS generation is one of the first processes that occurs before the mitochondrial membrane potential collapses, pro-apoptotic proteins are released, and caspases are activated [62].

The metastatic event is multi-factorial and multistep and includes such sequelae as cell invasion, access to the vasculature, cell migration through blood circulation, and colony formation at distant locations [63]. These activities can be inhibited by an anti-metastatic drug, resulting in less aggressive cancer cells. Invasion is essential for metastasis since motile cells must traverse the extracellular matrix and migrate into neighboring tissues. Using an in vitro scratch test to detect cell migration is a simple, low-cost, and well-established approach that is particularly well-suited for investigating the impact of cell-matrix and cell–cell interactions on cell migration [64]. In the current research, treatment with a sub-cytotoxic concentration of MN demonstrated a concentration-dependent decrease in MCF-7 cell migration. The current results are in good agreement with earlier studies, in which the *Murraya koenigii* plant-derived carbazole alkaloid girinimbine also showed very potent anti-metastasis activity in an in vivo model [65]. In cancer cells, cell migration needs the action of protease enzymes to digest the basement membrane and degrade connective tissue. Thus, this phase in the metastatic cascade may be a possible target for therapeutic candidates targeting cancer invasion [66]. Matrix metalloproteinases (MMPs), also known as matrixes, are enzymes that work in the extracellular environment of cells, degrading both matrix and non-matrix proteins [67]. MMPs contribute considerably to cancer growth by degrading the extracellular matrix and basement membrane, and they are the principal proteolytic enzymes involved in cancer invasion and metastasis. MMPs are strongly linked to breast cancer, and MMP-9 is the most often upregulated protein in breast cancer. MMP inhibitors have been reported to greatly boost apoptotic levels in human mammary epithelial cells when used in breast cancer therapy [68]. Molecular markers such as MMP-2 and MMP-9 can be utilized as reference indices to guide the diagnosis and treatment of breast cancer [69]. Our experimental findings demonstrate that the anti-invasion effect of MN in MCF-7 cells downregulates the activity of MMP-2 and -9 at both transcriptional and translational level.

## 4. Materials and Methods

### 4.1. Materials

We are grateful to Prof Mohd Aspollah Sukari from the Department of Chemistry, University Putra Malaysia, for providing us with the pure compound mahanimbine (MN) (Figure 1A). The isolation process and the spectra were previously published by Prof Mohd Aspollah Sukari [33,70,71,72]. The purity of the MN was confirmed by a single spot in TLC using pet ether/chloroform (1:1) as a mobile phase. All other materials were procured from a commercial supplier; details are provided in the Materials and Methods section.

### 4.2. Cell Culture and Cell Viability Measurement by MTT Assay

The human breast cancer cells (MCF-7 and MDA-MB-231) and non-cancer mammary cells (MCF-10A) were procured from ATCC (ATCC; Rockville, MD, USA). The cells were cultured as monolayers in RPMI-1640 media which was added with 10% (*v/v*) heat-inactivated FBS, penicillin (100 U/mL), streptomycin (100 µg/mL). All cell cultures were grown in a humidified incubator (Heraeus, GmbH, Germany) at 37 °C in 5% CO_2_.

The cell viability was measured by monitoring the conversion of 3-(4,5-dimethylthiazol-2-yl)-2,5-diphenyltetrazolium bromide (MTT, Sigma, St. Louis, MO, USA) dye according to the method described with slight modification [73]. The compound MN was dissolved in DMSO at 10 mg/100 µL concentration as a stock. All the triplicate experiments were included with 96-well plates, including cell-free control and vehicle control. Briefly, both the cells at a concentration of 15,000 cells per well were plated into 96-well plates (Nunc, Roskilde, Denmark). The number of cells was calculated using the preliminary assays and the doubling time of cells. The plated cells were incubated overnight for cell adhesion to plates. The next day, the media from the plates were discarded and filled with new serum-free media containing different concentrations of MN, and further incubated for 48 h. After incubation, 20 µL of 5 mg/mL of MTT reagent was added, and cells were incubated again for 4 h in the dark at 37 °C. DMSO was added to dissolve the formed crystals, and the absorbance of purple color formed was then measured using a microplate reader (BioTek Instruments, Inc, Winooski, VT, USA) at a wavelength of 570 nm with 630 nm as the reference wavelength. The percentage of cellular viability was calculated using Prism software with the appropriate controls taken into account [74].

### 4.3. Morphological Detection of Apoptosis

Initially, this study investigated the morphological changes that occurred during cell death in MCF-7 cells triggered by MN. Using a standard inverted microscope, the morphological aspect of treated cells was compared with that of the untreated control. After treatment, the morphological alterations of the cells were studied using a standard, inverted microscope. In this study, the IC_50_ value was employed for 12, 24, and 48 h. Meanwhile, the negative control consisted of cells that had not been treated.

The acridine orange (Sigma Chemical Co., St. Louis, MO, USA) and propidium iodide (Sigma Chemical Co., St. Louis, MO, USA) double-staining method was used to determine the different phases of apoptosis, such as early apoptosis, late apoptosis, and secondary necrosis. In a T25 flask, the MCF-7 cells were plated at a 1 × 10^6^ cells/mL density and incubated overnight. Then for 12, 24, and 48 h, the cells were treated with 14 µM MN. The treated cells were collected, centrifuged at 300× *g* for 10 min to remove the supernatant, and rewashed with PBS two times. The cells were then added with 1 µL of acridine orange (1 mg/mL) and propidium iodide (1 mg/mL) (1:1) and kept in the dark for 10 min at room temperature. The cell suspension was dropped into a glass slide, covered by a coverslip, and examined under a fluorescence microscope (GXM-L3201, GT Vision, Suffolk, UK). The total number of viable/dead cells in different categories was determined in >200 cells. In this case, the intercalating fluorochromes AO and PI were selective for nucleic acids and exhibited green and orange fluorescence, respectively, when they were coupled to DNA; only AO can cross the plasma membrane of live and early apoptotic cells. The criteria for identifying different cell populations was carried out according to the method described earlier [75].

### 4.4. Determination of Intracellular ROS Generation by DCFDA

DCFDA assay was carried out to determine a cell’s redox state. In principle, the presence of intracellular ROS will initiate deacetylation of DCFDA esterases, then oxidize to form the fluorescent product 2′,7′-dichlorofluorescein. MCF-7 cells were plated into six-well plates and incubated for 24 h for attachment. On day two, the cells were treated with 14 µM MN for 12, 24, and 48 h. One micromolar hydrogen peroxide (H_2_O_2_) was used as a positive control for ROS generation. After treatment, the cells were collected, centrifuged at 300× *g* for 10 min to remove the supernatant, and the cells were washed again with PBS two times. The cells were then added with 20 μM DCFDA (Sigma, St Louis, MO, USA) for 15 min in the dark, followed by washing with PBS to remove the unbound dye, and were visualized with a fluorescent microscope (GXM-L 3201, GT Vision, UK). Quantitative analysis of fluorescence intensity in cells was performed using ImageJ software (http://rsbweb.nih.gov/ij/; accessed on 20 Decembar 2021).

### 4.5. Determination of Mitochondrial Membrane Potential

Disruption of mitochondrial membrane potential (Δψm) has a pivotal role in apoptosis-mediated cell death, predominantly mediated via intrinsic pathways. In this study, JC-1 dye (Cayman chemicals, Ann Arbor, MI, USA) was used to determine the loss of Δψm. Treatment of MCF-7 cells was carried out using 14 µM MN for 12 and 24 h. After treatment, the washed cells were exposed to JC-1 dye (10 μg/mL) for 30 min at 37 °C. Subsequently, the cells were rinsed and incubated with 200 μL of assay buffer; cells were then observed under a fluorescent microscope (GXM-L3201, GT Vision, Suffolk UK).

### 4.6. Determination of Caspase Activity

To determine the expression of initiator and executive caspases associated with apoptosis, Caspase-Glo-9, Caspase-Glo-3/7, and Caspase-Glo-8 assay reagents (Promega, Madison, WI, USA) were used according to the protocol by the manufacturer. Since the caspase-3 gene was functionally deleted in MCF-7 cells, there was no caspase-3 activity present. As a result, caspase-7 activity was what was being measured in this experiment. Briefly, MCF-7 cells were treated with 14 µM of MN for 12, 24, and 48 h in opaque 96-well plates in triplicate. At the end of the experiment, the assay buffer was added with the plates and incubated for 1 h. The luminescence produced was measured using a plate reader (Infinite 200 PRO Microplate Reader, Tecan, GmbH, Grödig, Austria).

### 4.7. Wound Healing Assays

A wound-healing assay was carried out as described earlier [76]. MCF-7 cells were plated into 6-well plates at a 1 × 10^6^ cells/mL density for 24 h to attach. The next day, using a 200 µL pipette tip, a vertical scratch was made in the cells’ confluent monolayer. Floating cells and the debris were then removed from the wells using sterile PBS and then incubated with 2 mL of low serum media (1% FBS) containing a sub-cytotoxic concentration of (10 µM) MN that was added. Cell migration was monitored for the next 12, 24, and 48 h using phase-contrast microscope. Images were captured at 20 X magnification. Wound healing distance was measured using ImageJ software and compared between treatment and control groups.

### 4.8. Gene Expression Analysis

SYBR-green RT-qPCR was used to determine the expression level of Bax, Bcl-2, MMP-2, and MMP-9 in the treated MCF-7 cells. In brief, MCF-7 cells were treated with MN for 12, 24, and 48 h in a T25 flask at a 1 × 10^6^ cells/mL density. According to the instruction, total RNA was extracted from the treated and control cells using RNA-sure Fusion RNA mini kit (Genetix Biotech, New Delhi, India). The quantity and purity of extracted RNA were measured using Nanodrop 2000 spectrophotometer (Thermo Scientific). Subsequently, complementary DNA was prepared using 10 µL (approximately 2 μg) of RNA using cDNA reverse transcription kit (Applied Biosystems, Forster City, CA, USA) with incubation at room temperature for 10 min. Then RT was conducted for 120 min at 37 °C and 5 min at 85 °C. PCR amplification was performed by using target gene primers with β-actin as an internal control. The PCR primers sequence used were for β-actin, sense primer 5′-TCCCTGGAGAAGAGCTACG-3′ and antisense primer 5′-GTAGTTTCGTGGATGCCACA -3′; for Bax, sense primer 5′-ACGAACTGGACAGTAACATGGAG-3′ and antisense primer 5′-CTTCTTCCAGATGGTGAGTGA-3′; for Bcl-2, sense primer 5′-CTCGTCGCTACCGTCGTGACTTCG-3′ and antisense primer 5′-ACCCCATCCCTGAAGAGTTCC-3′; for MMP-2, sense primer 5′-GATACCCCTTTGACGGTAAGGA-3′ and antisense primer 5′-CCTTCTCCCAAGGTCCATAGC-3′; for MMP-9, sense primer 5′-GGGACGCAGACATCGTCATC-3′ and antisense primer 5′-TCGTCATCGTCGAAATGGGC-3′ [77,78]. The qPCR program included 95 °C for 3 min, followed by 40 cycles of 12 sec at 95 °C denaturation, 30 sec at 60 °C annealing, 72 °C/30 sec for an extension. The cycle threshold (CT) values were determined by automated threshold analysis. The findings of the real-time PCR experiments were then quantified using the delta-delta Ct (Ct) technique for comparative threshold cycle (Ct) quantification of gene expression.

### 4.9. Protein Expression Analysis

The levels of protein expression were assessed using colorimetric techniques using Bcl-2 (Cat# HB0040), Bax (Cat# HB0040), MMP-2 (Cat#HM0311), and MMP-9 (Cat# HM0336) ELISA kit (Neo Biolabs, F Cambridge, MA, USA) according to the manufacturer’s instructions. To summarize, a mammalian Protein Extraction Reagent (Thermo Fisher, Waltham, MA, USA) was used to extract the proteins from the cells. Bradford protein quantitation assay was used to measure the protein concentration from the cell lysate. An equal amount of proteins was later coupled selectively to the primary antibody, and proteins were identified using a secondary antibody linked with horseradish peroxidase (HRP). The secondary antibody coupled to HRP enabled sensitive colorimetric absorption was measured by a microplate reader (Bio-Tek Instruments, Highland Park, TX, USA) at 450 nm and the reference wavelength of 630 nm.

### 4.10. Statistical Analysis

Statistical analysis was performed using Graphpad Prism (GraphPad Software Version 8, San Diego, CA, USA). All values were measured as mean ± SD. Values of *p* < 0.05 and *p* < 0.01 were considered significant after one-way ANOVA.

## 5. Conclusions

In conclusion, MN demonstrated cytotoxic and anti-migratory activities against human breast cancer MCF-7 cells by increasing cancer cell apoptosis. According to the present data, it can be clearly seen that MN inhibits cancer cell growth through programmed cell death. In the presence of MN, MCF-7 cells went into apoptosis, producing cell death signals that controlled Δψm by downregulating Bcl2 and upregulating Bax. The cell death was found to be augmented with the significant release of ROS. The cell death was associated with caspase-9 and -7 enzyme release with the absence of caspase-8 trigger, suggesting that the apoptosis had happened via the mitochondrial pathway. In addition, MN decreased cancer cell migration via reduction in MMP-2 and -9. In summary, the results presented here suggest that MN is a significant compound from a natural product to be evaluated further for anti-breast cancer drug discovery as a lead compound.

## Figures and Tables

**Figure 1 molecules-27-00971-f001:**
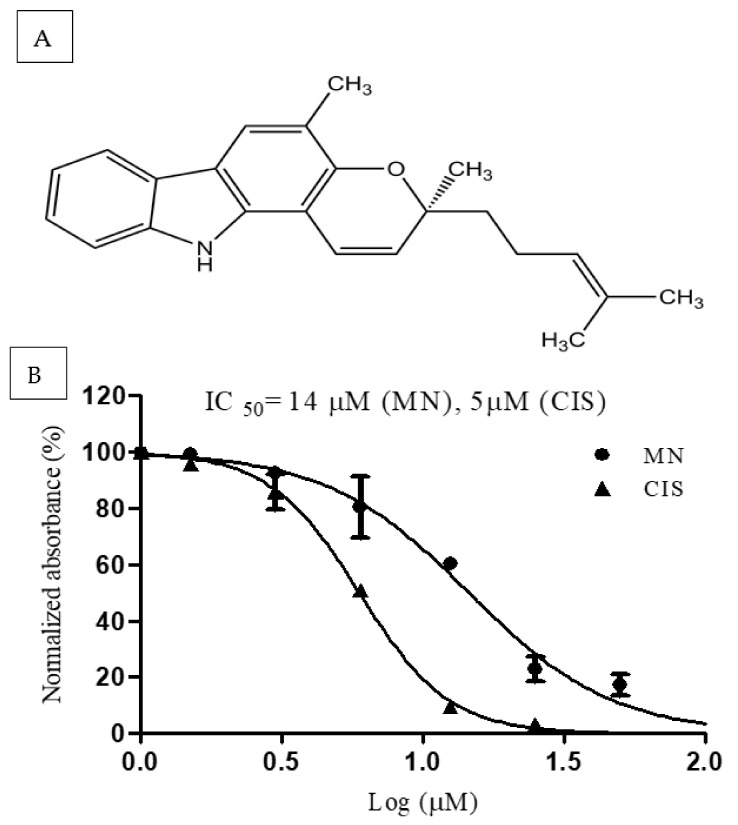
Chemical structure and effect of mahanimbine (MN) on the proliferation of breast cancer cells. (**A**) Chemical structure of mahanimbine. (**B**) Antiproliferative effect of MN and cisplatin (CIS) on MCF-7 cells at various dose concentrations for 48 h. All the experiments were carried in triplicate, and data are expressed as mean ± SD values.

**Figure 2 molecules-27-00971-f002:**
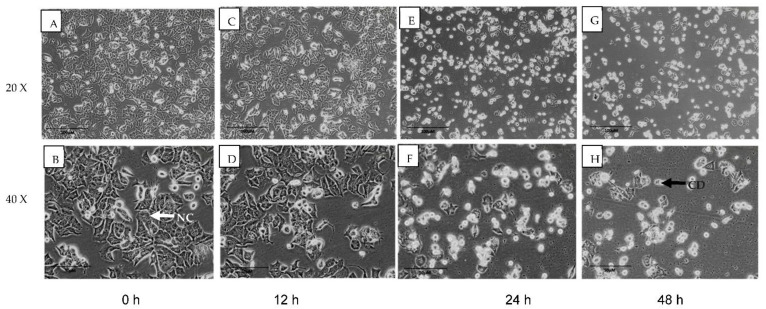
Normal contrast microscopic examination of MCF-7 cells. Cells treated with IC_50_ of MN (14 µM) in a time-dependent manner. Treated cells morphology compared well with untreated control cells. (**A**,**B**) represent 0 h; (**C**,**D**) represent 12 h; (**E**,**F**) represent 24 h; (**G**,**H**) represent 48 h treatment with MN. NC is normal cell; CD is cell debris.

**Figure 3 molecules-27-00971-f003:**
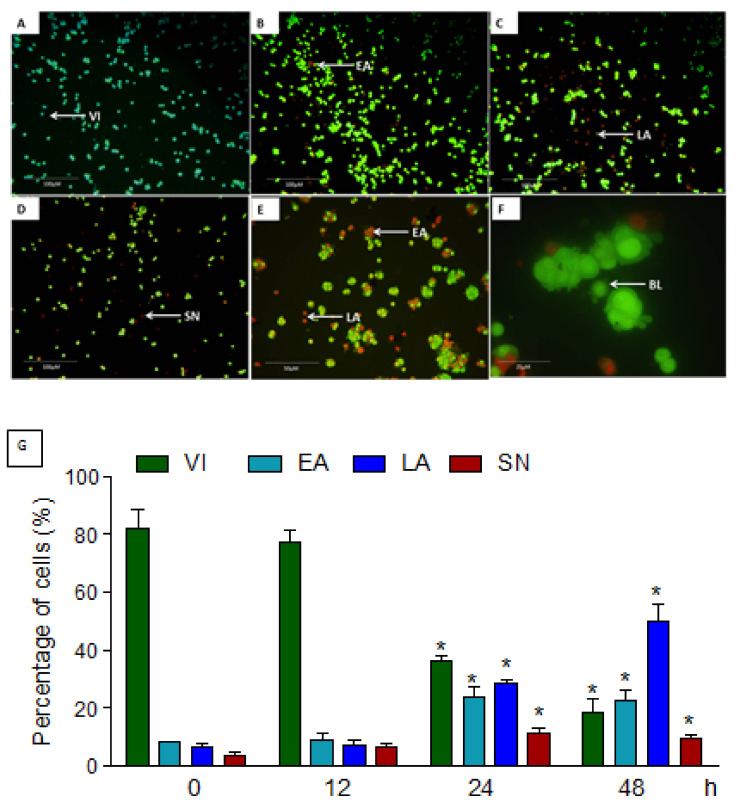
MCF-7 cells with AO/PI double staining. The cells were treated for 12, 24, and 48 h with 14 µM MN. (**A**) Untreated cells, (**B**) cells treated for 12 h, (**C**) cells treated for 24 h, (**D**) cells treated for 48 h. Cell treated with MN had shown early apoptosis sign with bright green color, which rose to late apoptosis with orange color at 24 h. At 48 h, both late apoptosis (orange) and secondary necrosis (red) cells were observed. Apoptosis features such as fragmented DNA, chromatin condensation, and membrane blebbing were observed at 48 h (**E**,**F**). (**G**) Treatment of MCF-7 cells with MN improved the percentages of viable early and late apoptosis and secondary necrotic cells significantly (* *p* < 0.05) in a time-dependent manner. VI: viable cells, EA: early apoptosis, LA: late apoptosis, SN: necrosis.

**Figure 4 molecules-27-00971-f004:**
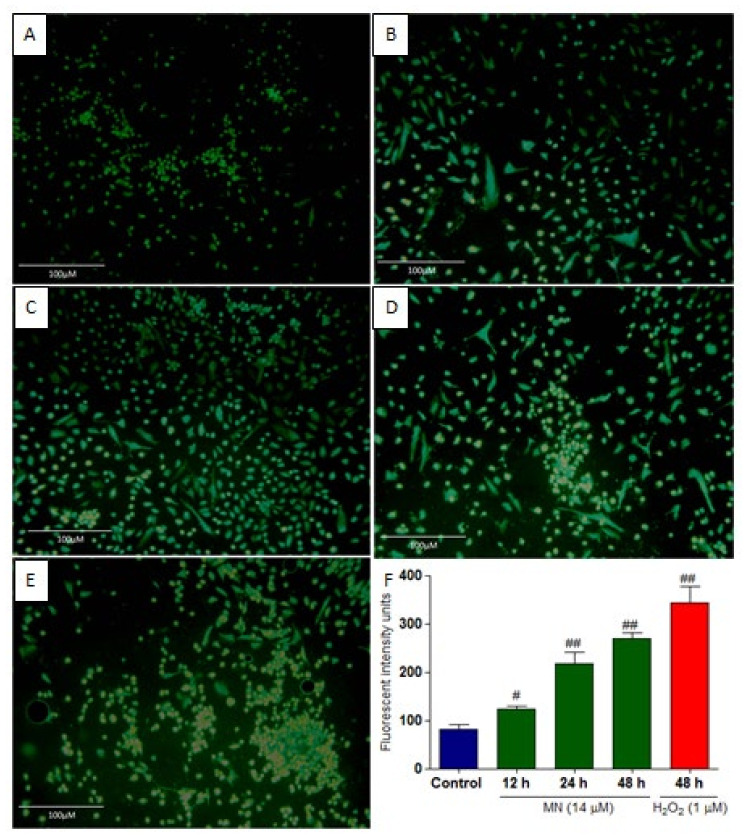
Fluorescence microscope images of DCFDA. Cells were treated with 14 µM of MN for 12 (**B**), 24 (**C**), and 48 (**D**) h. As a positive control for ROS production, cells were treated with 1 µM hydrogen peroxide (H_2_O_2_) (**E**). Results were compared with untreated control (**A**). Fluorescence microscopy was used to view the cells, and pictures were acquired at a magnification of 20×. (**F**) Using ImageJ software, three distinct fields were randomly counted for positive green cells, and the average fluorescence intensity of each concentration of MN was presented. All data are represented as mean ± SD. ^#^
*p* < 0.05, ^##^
*p* < 0.01 vs. control *n* = 3.

**Figure 5 molecules-27-00971-f005:**
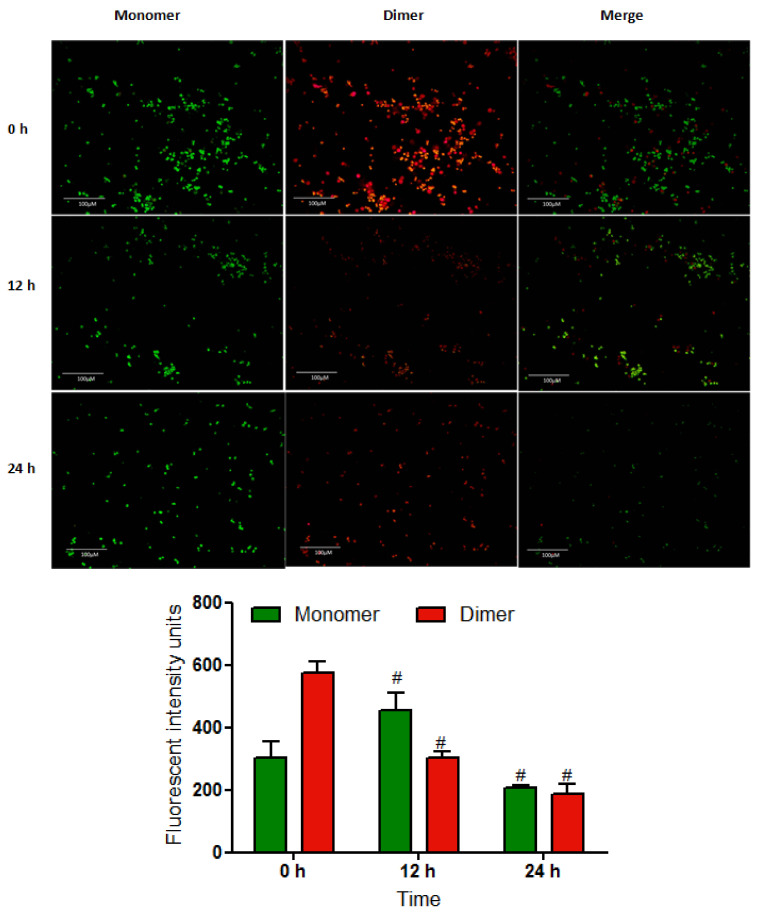
Effect of MN on the Δψm in MCF-7 cells stained with JC-1 dye. Pictures shown are JC-1 monomer, J-aggregates dimer (red fluorescence), and merged images. As the pictures show, the accumulation of JC-1 dye in the mitochondria matrix generates red fluorescence. As the Δψm decreased, JC-1 could not aggregate in the matrix, causing JC-1 to remain as a monomer, resulting in green fluorescence. The quantified data showed that the red fluorescent decreased significantly as time increased. All data are represented as mean ± SD. ^#^
*p* < 0.05 vs. control *n* = 3.

**Figure 6 molecules-27-00971-f006:**
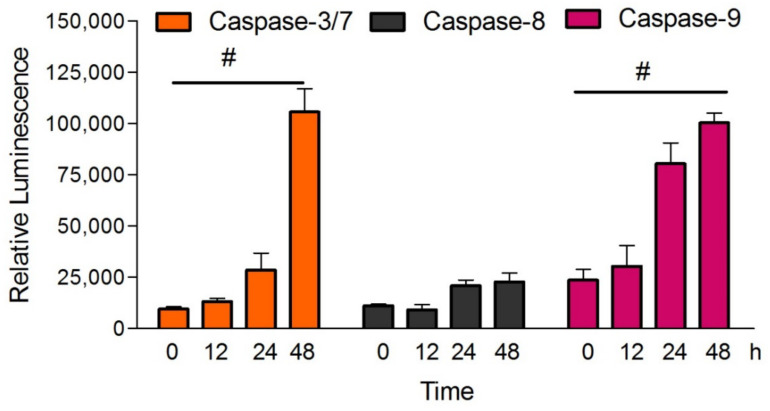
Caspase-3/7,-8, and -9 relative luminescence expression in MCF-7 cells treated with MN for 12, 24, and 48 h. Each independent experiment employed triplicates from each treatment group. All data are represented as mean ± SD. ^#^
*p* < 0.05 vs. control *n* = 3.

**Figure 7 molecules-27-00971-f007:**
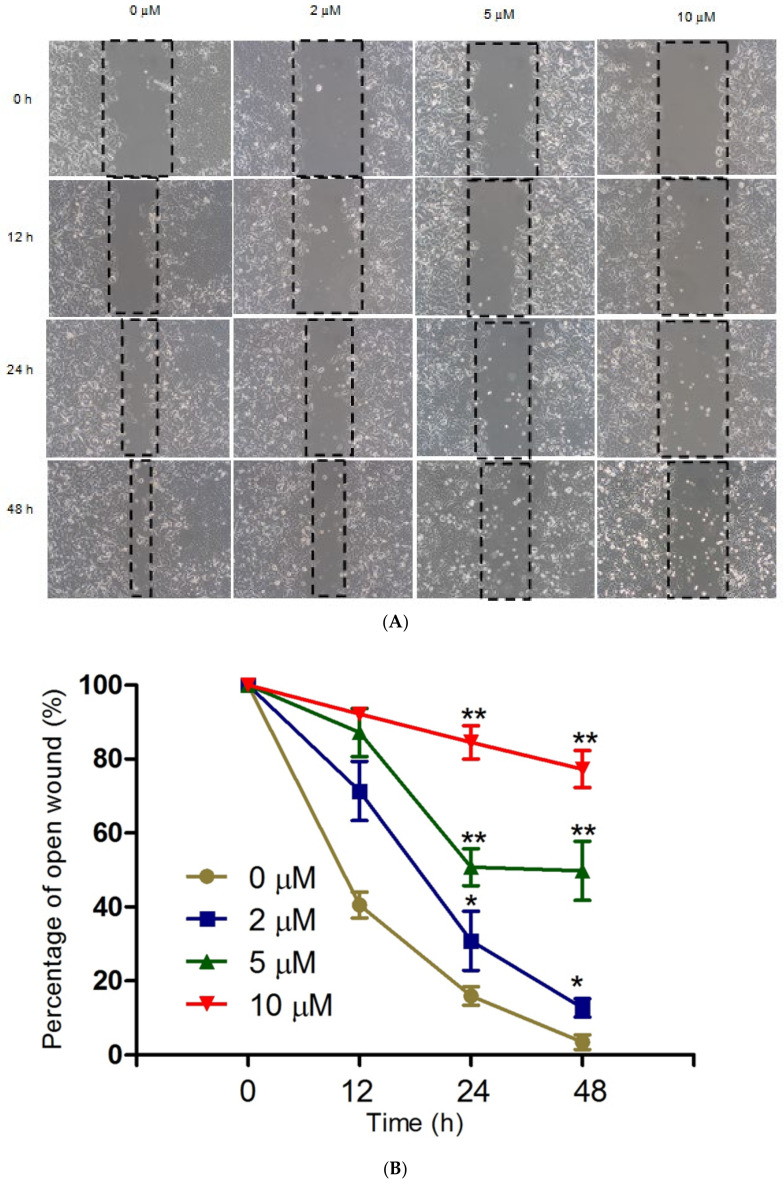
Anti-migratory ability of MN was determined by wound healing assay in MCF-7 cells. (**A**) Cells were scratched using 200 pipette tips and treated with 2, 5, and 10 µM (sub-cytotoxic concentrations) of MN for 12, 24, and 48 h. Photographs were taken at each time point at the same scratch site at 20× magnification. (**B**) The line graph shows the quantified cell migration measure by ImageJ software. Each independent experiment employed triplicates from each treatment group. * *p* < 0.05 and ** *p* < 0.01 vs. control *n* = 3.

**Figure 8 molecules-27-00971-f008:**
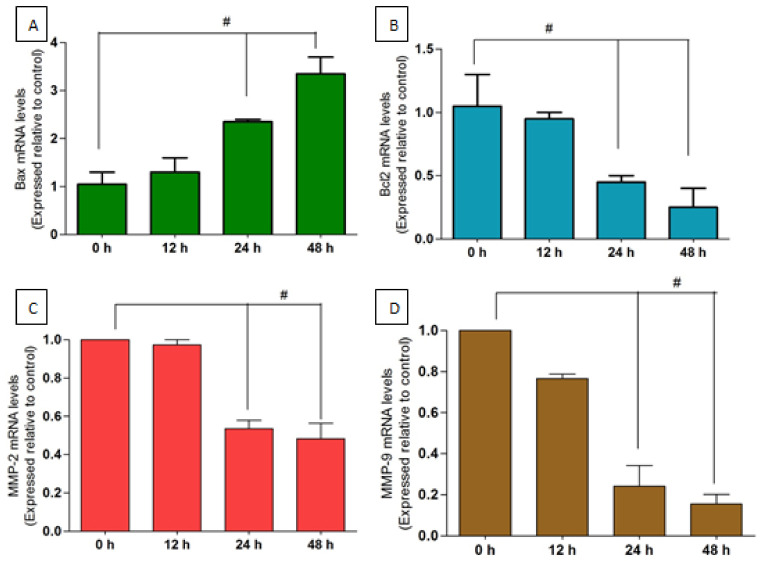
The effect of MN on Bax, Bcl2, MMP-2, and MMP-9 mRNA levels in MCF-7 cells. (**A**,**B**) Cells were treated with 14 µM MN for 12, 24, and 48 h, and Bax and Bcl2 mRNA expressions levels were determined by RT-PCR with β-actin as an internal control. (**C**,**D**) Cells were treated with 10 µM (sub-cytotoxic concentration) MN for 12, RT-PCR determined 24 and 48 h and MMP-2 and MMP-9 mRNA expressions levels with β-actin as an internal control. All data are represented as mean ± SD. ^#^
*p* < 0.05 vs. 0 h. *n* = 3.

**Figure 9 molecules-27-00971-f009:**
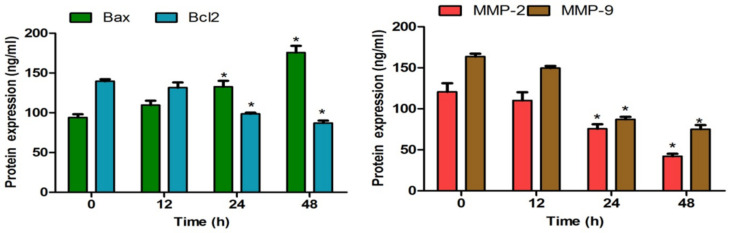
Bax, Bcl-2, and MMP-2 and -9 protein expression in MCF-7 cells treated with MN for 12, 24, and 48 h evaluated by ELISA. An amount of 14 µM MN was used for Bax and Bcl2 expression determination; meanwhile, 10 µM (sub-cytotoxic concentration) of MN was used for MMP-2 and -9 expressions. All data are represented as mean ± SD. * *p* < 0.05 vs. 0 h. *n* = 3.

## Data Availability

The author confirms that the data supporting the findings of this study are available for readers.

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
