# Peer review of "Cytotoxicity of Mahanimbine from Curry Leaves in Human Breast Cancer Cells (MCF-7) via Mitochondrial Apoptosis and Anti-Angiogenesis"

_molecules, 2022, doi:10.3390/molecules27030971_

Round 1

Reviewer 1 Report

The manuscript „Cytotoxicity of mahanimbine from curry leaves in human breast cancer cells (MCF-7) via mitochondrial apoptosis and anti-angiogenesis“ by Yahya Hasan Hobani reports on the comprehensive investigations of the mechanism of anticancer action exerted by the phytochemical mahanimbine against MCF-7 cells. The experimental design covers investigation of cytotoxicity, induction of apoptosis, insights into the effect on apoptosis-related proteins/enzymes, impact on mitochondrial membrane potential and cell migration. Generally, the topic of the manuscript very well fits the scope of the special issue focusing on the characterization of bioactive compounds obtained from natural sources, e.g., phytochemicals, and its application as anticancer compound. In order to improve the quality of the manuscript further, I wish to make the following comments:

Major aspects:

@ introduction: The introduction would benefit from the embedding of more recent work on breast cancer and its treatment such as https://doi.org/10.3389/fphar.2020.632079 from 2021, https://doi.org/10.1016/S0140-6736(20)32381-3 from 2021, and https://www.mdpi.com/2072-6694/13/17/4287 from 2021 – thus emphasizing the importance of the current study.

@ Figure 2: Please revise the presentation of the figure: What does 20 X and 40 X mean – is it either with or without treatment? Insert space between digit and unit (also, e.g., lines 103, 118, 142, 148, 157, Figure 7A, 193, 198, 369, 380, 412 – listing not completed). Please remove the formatting symbols (also Figures 3, 4, 5).

@ Figure 3G: After 24 and 48 h, EA and especially LA increase. Conspicuously, there are hardly any distinct effects after treatment for 12 h. The author is kindly asked to discuss on this kind of “time-lag”.

@ Figure 6: Please be consistent with the usage of space and hyphen for labelling the caspases. – The symbol of significance should be # in this figure. Please provide information about the number of replicates (n=?) and the way of data presentation (mean ± SD?). – It is very obvious that Caspase-8 is not affected by treatment with MN. The author should discuss on that striking finding in the section 2.5 of the manuscript. How about adducing a control within the experiments when investigating the induction of caspase?

@ Figures 8 and 9: Using the same pattern (or rather using same colors) for each protein would allow for better comparison between these figures and would make both figures more reader-friendly.

@ MCF-7: The current study was performed based on the earlier investigations cited in references [24,25]. However, it would be of interest to know the effects of MN on other cell lines, especially on triple negative breast cancer cell lines (estrogen negative, progesterone negative, human epidermal growth factor 2 negative) since MCF-7 is not classified as triple negative. The author is kindly asked to discuss on that (while experimental evidence using a triple negative cell line would be supportive). In contrast, the cytotoxicity of MN expressed as IC50 with respect to non-cancer cell line MCF-10A is just 2-fold lower, limiting the use of MN. But it probably might be active against triple negative breast cancer cell lines, thus expanding the value of the compound.

@ Discussion: At same parts, this paragraphs rather resembles an enumeration of the results. It is highly suggested to prune the text and to concentrate on the main findings.

@ supplementary material: The document does not contain any cover indicating the belonging to the current paper and “file1” does not represent a proper labelling. However, it is not necessary to provide these data as supporting material, because lines 302-303: “The isolation process and the spectra have been previously published by Prof Mohd Aspollah Sukari [21, 44-46]; its details have been provided as supplementary file1.” It is already published and can be looked up in literature.

Although there will be a close editing by the MDPI publisher, the author is kindly asked to correct formal/formatting issues, such as use of space between digits and units, consistent labelling of figures in the main text (case shift, space), avoiding of repletion (show), abbreviation of proteins (with/without space, hyphen), sometimes different fonts, consistent representation style of the references.

Minor aspects:

@ line 38: “Cancer drug research has several different targets. One of these is reactive oxygen species (ROS)” This statement should be revised: Are ROS really a drug target? It should rather be said that induction of ROS is a strategy of anticancer treatment.

@ line 69: “Mahanimbine (MN) is a carbazole alkaloid and is present in the leaves of this plant” The author is kindly asked to add a reference to Figure 1 A displaying the chemical structure of MN.

@ line 70: “Available data show mahanimbine's high pharmacological potential…” The author already gives a nice overview of the different pharmacological properties of MN, however, it is suggested to add the aspects reported in the publications https://www.mdpi.com/2076-3425/12/1/12/htm from 2022 and https://doi.org/10.1002/biof.1333 from 2017.

@ line 82: “…while the IC50 of cisplatin was 5 ± 0.08 µM” The value of the reference is in good agreement with literature. Citation of a reference, e.g. https://doi.org/10.1039/C9DT03330K, would pronounce the comparability of the current study.

@ line 82/308: “The normal mammary cells (MCF-10A)” The usage of “normal” does not seem that precise. It is recommended to term MCF-10A a “non-cancer” cell line.

@  lines 82-83: The reporting of the cytotoxicity vales should be consistent regarding significant figures.

@ Figure 1: Please revise the caption. Actually, Figure 1 A does not represent “Effect of Mahanimbine (MN) on the proliferation of breast cancer cells.”

@ line 98: “…MCF-7 cells treated with MN for 12, 24, and 48 h” Please add the concentration of treatment.

@ line 125: “the MN's ability” Please revise this phrasing.

@ line 126: Please introduce the abbreviation DCFDA.

@ Figure 4: Description of (A) is missing in the caption.

@ line 144: “1 micro M” Please consistently use Greek symbol.

@ line 166: “at the same field and magnification to compare the results” Please add a note with which the results are compared with.

@ section 2.6: Please add a short information on the meaning of the migration respectively the anti-migration.

@ line 204: “Breast cancer is one of the deadliest cancer among women…” Is that really true?

@ line 249: “causes a catastrophic modification” Please revise this phrasing.

@ e.g., lines 339/341/358/384/395: Please revise the use of “X”.

Author Response

every points from the reviewer comments were properly replied in the attached document

Reviewer 2 Report

Manuscript ID: molecules 1548113

Title: Cytotoxicity of mahanimbine from curry leaves in human breast cancer cells (MCF-7) via mitochondrial apoptosis and anti-angiogenesis  

The manuscript presents the biological activity of carbazole alkaloid from Murraya koenigii leaves. The research was well designed; however, I consider that the authors should reconsider the novelty and contribution of this work to the knowledge because there are several papers on the anticancer activity from Murraya koenigii extracts. Below, I indicate some related works that the authors omitted in this topic:  

Study of physicochemical, nutritional, and anticancer activity of Murraya Koenigii extract for its fermented beverage. https://doi.org/10.1111/jfpp.16137

Phytochemical Analysis and Evaluation of Antioxidant and Biological Activities of Extracts from Three Clauseneae Plants in Northern Thailand. https://doi.org/10.3390/plants10010117

Mukonal exerts anticancer effects on the human breast cancer cells by inducing autophagy and apoptosis and inhibits the tumor growth in vivo. https://doi.org/10.1186/s13568-020-01074-8

Anticancer Activity of Mukonal Against Human Laryngeal Cancer Cells Involves Apoptosis, Cell Cycle Arrest, and Inhibition of PI3K/AKT and MEK/ERK Signalling Pathways. DOI: 10.12659/MSM.910702

Carbazole alkaloids from Murraya koenigii trigger apoptosis and autophagic flux inhibition in human oral squamous cell carcinoma cells. https://doi.org/10.1007/s11418-016-1045-6

Cytotoxicity and Proteasome Inhibition by Alkaloid Extract from Murraya koenigii Leaves in Breast Cancer Cells—Molecular Docking Studies. https://doi.org/10.1089/jmf.2016.3767

In Vivo Inhibition of Proteasome Activity and Tumour Growth by Murraya koenigii Leaf Extract in Breast Cancer Xenografts and by Its Active Flavonoids in Breast Cancer Cells. DOI: 10.2174/1871520616666160520112210

Antiproliferative and caspase-mediated apoptosis inducing effects of Murraya koenigii seeds against cancer cells. https://doi.org/10.1016/j.sajb.2020.05.021

Preclinical Development of Mahanine-Enriched Fraction from Indian Spice Murraya koenigii for the Management of Cancer: Efficacy, Temperature/pH stability, Pharmacokinetics, Acute and Chronic Toxicity (14-180 Days) Studies. https://doi.org/10.1155/2020/4638132

Phytochemical screening and In vitro Cytotoxic activity of Hexane extract of Temurui (Murraya koenigii (Linn.) Spreng) leaves against Human Cervical Cancer (HeLa) cell line. doi:10.1088/1757-899X/523/1/012018

Besides, the discussion section is poor and does not reflect the impact of the results.  

Author Response

(The authors gave the same response as above.)

Reviewer 3 Report

Dear Author

Interesting paper well written missing only a few things

Line 79 refers figure 1B without having indicated figure 1A

Figure 2: it is easy to find CD cell debris, but NC normal cells the arrow cannot be easily identified. The images during 12, 24 and 48h are always on the same spot as in the 0h?

Line 124: «trigger apoptosis is the accumulation of ROS». ROS triggers apoptosis but what means this for the normal cells? Could this be considered side effect of MN? This should be taken into account in the Discussion.

Line 126: DCFDA it should be written first

Discussion: it is well written having into account the biological activity of all the parameters studied, but a comparison with other natural products, mainly terpenoids is lacking. It would be interesting to compare the biological effect with other natural compounds and to discuss side effects that these compounds may also have.

Author Response

(The authors gave the same response as above.)

Round 2

Reviewer 1 Report

Review on the revised version of the manuscript „Cytotoxicity of mahanimbine from curry leaves in human breast cancer cells (MCF-7) via mitochondrial apoptosis and anti-angiogenesis“ by Yahya Hasan Hobani submitted to consider for publication as Article in the journal “Molecules”.

The author did a very diligent job to act on every concern and suggestion mentioned before. Every aspect was addressed, corrected or optimized. Herewith, I wish to focus on two particular comments.

@ comment 6, 2-fold lower vs. 2-fold higher: I am sorry for this inconvenience. Apologies! The author is right.

@ comment 15, significant digits: “The MN inhibited the MCF-7 cells and MDA-MB-231 cells with an IC50 of 14 ± 0.2 and 21.5 ± 0.8 μM respectively, while the IC50 of cisplatin was 5 ± 0.08 μM in an agreement with an earlier study [40]. The non-cancer mammary cells (MCF-10A) had shown an IC50 of 30.5 ± 1.4 μM on MN treatment.” The reporting of the standard deviation should not be more precise (regarding significant digits) than the mean value itself, i.e. 14.0 ± 0.2 μM and 5.00 ± 0.08 μM or 14 ± 0 μM and 5 ± 0 μM. Please adjust the values.

Moreover, I wish to praise the comprehensive embedding of previous work on carbazole alkaloids in the current study. This gives a nice overview.

In conclusion, I suggest further proceeding of the manuscript for publication. All the best!

Reviewer 2 Report

The authors have addressed the comments made to the manuscript and the paper is accepted.